# Preoperative Nutrition Intervention in Patients Undergoing Resection for Upper Gastrointestinal Cancer: Results from the Multi-Centre NOURISH Point Prevalence Study

**DOI:** 10.3390/nu13093205

**Published:** 2021-09-15

**Authors:** Irene Deftereos, Justin M.-C. Yeung, Janan Arslan, Vanessa M. Carter, Elizabeth Isenring, Nicole Kiss

**Affiliations:** 1Department of Surgery, Western Precinct, Melbourne Medical School, The University of Melbourne, St Albans, VIC 3021, Australia; justin.yeung@unimelb.edu.au (J.M.-C.Y.); janan.arslan@unimelb.edu.au (J.A.); 2Department of Nutrition and Dietetics, Western Health, Footscray, VIC 3011, Australia; vanessa.carter@wh.org.au; 3Department of Colorectal Surgery, Western Health, Footscray, VIC 3011, Australia; 4Western Health Chronic Disease Alliance, Western Health, Footscray, VIC 3011, Australia; 5Faculty of Health Sciences and Medicine, Bond University, Robina, QLD 4226, Australia; lisenrin@bond.edu.au; 6Department of Nutrition and Dietetics, Princess Alexandra Hospital, Brisbane, QLD 4102, Australia; 7Institute for Physical Activity and Nutrition, Deakin University, Geelong, VIC 3220, Australia; nicole.kiss@deakin.edu.au; 8Allied Health Research, Peter MacCallum Cancer Centre, Melbourne, VIC 3000, Australia

**Keywords:** gastrointestinal cancer, preoperative, dietitian, nutrition support, malnutrition

## Abstract

Background: Preoperative nutrition intervention is recommended prior to upper gastrointestinal (UGI) cancer resection; however, there is limited understanding of interventions received in current clinical practice. This study investigated type and frequency of preoperative dietetics intervention and nutrition support received and clinical and demographic factors associated with receipt of intervention. Associations between intervention and preoperative weight loss, surgical length of stay (LOS), and complications were also investigated. Methods: The NOURISH Point Prevalence Study was conducted between September 2019 and May 2020 across 27 Australian tertiary centres. Subjective global assessment and weight were performed within 7 days of admission. Patients reported on preoperative dietetics and nutrition intervention, and surgical LOS and complications were recorded. Results: Two-hundred patients participated (59% male, mean (standard deviation) age 67 (10)). Sixty percent had seen a dietitian preoperatively, whilst 50% were receiving nutrition support (92% oral nutrition support (ONS)). Patients undergoing pancreatic surgery were less likely to receive dietetics intervention and nutrition support than oesophageal or gastric surgeries (*p* < 0.001 and *p* = 0.029, respectively). Neoadjuvant therapy (*p* = 0.003) and malnutrition (*p* = 0.046) remained independently associated with receiving dietetics intervention; however, 31.3% of malnourished patients had not seen a dietitian. Patients who received ≥3 dietetics appointments had lower mean (SD) percentage weight loss at the 1-month preoperative timeframe compared with patients who received 0–2 appointments (1.2 (2.0) vs. 3.1 (3.3), *p* = 0.001). Patients who received ONS for >2 weeks had lower mean (SD) percentage weight loss than those who did not (1.2 (1.8) vs. 2.9 (3.4), *p* = 0.001). In malnourished patients, total dietetics appointments ≥3 was independently associated with reduced surgical complications (odds ratio 0.2, 95% confidence interval (CI) 0.1, 0.9, *p* = 0.04), and ONS >2 weeks was associated with reduced LOS (regression coefficient −7.3, 95% CI −14.3, −0.3, *p* = 0.04). Conclusions: Despite recommendations, there are low rates of preoperative dietetics consultation and nutrition support in this population, which are associated with increased preoperative weight loss and risk of increased LOS and complications in malnourished patients. The results of this study provide insights into evidence–practice gaps for improvement and data to support further research regarding optimal methods of preoperative nutrition support.

## 1. Introduction

Curative intent surgeries for upper gastrointestinal (UGI) cancer including oesophagectomy, gastrectomy, and pancreatectomy carry a high postoperative morbidity risk [1]. Patients with UGI cancer can experience high rates of preoperative malnutrition, which has been associated with poor surgical outcomes, including increased length of stay (LOS) and complications [2]. As such, preoperative nutritional assessment and intervention are recommended by evidence-based surgical and oncology guidelines in order to reduce postoperative metabolic stress and improve surgical outcomes [3,4]. Although there are no evidence-based guidelines specific to UGI cancers, the European Society of Clinical Nutrition and Metabolism (ESPEN) guidelines consider patients undergoing UGI cancer surgery to be one of the highest priority groups for perioperative nutritional intervention and support [4].

Although preoperative nutritional intervention is widely recognised as a key aspect of surgical preparation, translation of recommendations into practice is difficult to achieve [5]. Surveys of gastrointestinal surgeons and dietitians report low rates of preoperative nutrition assessment and support, despite clinicians recognizing the importance of nutritional optimisation [6,7]. To move towards improved practice, an understanding of the current nutrition interventions received by patients in everyday clinical practice is required. Establishing minimum data sets to assist in the development of standardised nutrition protocols have been previously suggested in oncology populations [8]. Benchmarking current practice can assist in targeting key areas for immediate improvement and education, as well as further research to inform the development of UGI specific evidence-based guidelines. Instrumental observational studies in other clinical areas, including the intensive care International Nutrition Survey [9] and the Nutrition Care Day Survey of hospitalized patients [10], have led to the development of further research and subsequent practice change. More recently, the INFORM study in foregut tumours demonstrated ongoing evidence–practice gaps and highlighted key areas of the nutrition care process that require improvement [11]. However, no large multi-centre studies have been conducted to investigate and benchmark current perioperative nutrition practice and associated outcomes in patients undergoing curative intent UGI cancer resection.

The multi-centre Nutritional Outcomes of Patients Undergoing Resection for Upper Gastrointestinal Cancer in Australian Hospitals study (The NOURISH Point Prevalence Study) was therefore initiated to investigate the nutritional status and perioperative nutritional interventions received by patients undergoing curative intent surgery for UGI cancer [12]. Outcomes of the NOURISH study, including detailed analysis of nutritional status, site-specific nutritional practices, and perioperative nutritional intervention will be reported in a series of sub-studies. This is the second sub-study, and the primary aims were to determine the type and frequency of preoperative dietetics intervention and nutrition support received. Secondary aims were to determine factors associated with receipt of preoperative dietetics intervention and nutrition support, and to investigate associations between intervention and preoperative weight loss and surgical outcomes (complications and LOS).

## 2. Methods

### 2.1. Study Design and Population

The NOURISH Point Prevalence Study was conducted between 2 September 2019 and 30 June 2020. Twenty-seven tertiary hospitals across six Australian states participated. Ethics approval was obtained from The Peter MacCallum Cancer Centre Ethics Committee prior to commencement (LNR/51107/PMCC-2019). Further details of study design, participating sites, and methods are reported in the previously published study protocol [12]. Verbal consent was provided by participants and all patients received the standard dietetics care of the participating health service. Eligible participants were ≥18 years of age; a hospital inpatient having received curative intent surgery for UGI cancer, including gastrectomy (total, subtotal, distal, partial), pancreatectomy (total, distal, partial, pancreatico-duodenectomy), oesophagectomy, or gastro-oesophagectomy; able to consent to participation by English language communication or with the presence of an interpreter; and had received assessment of nutritional status with the Subjective Global Assessment (SGA) by a dietitian within seven days of surgery. Participants were ineligible to participate if they had received palliative surgery or non-oncological UGI surgery; were unable to participate in SGA; were unable to provide consent, including if they were on intravenous opioids at time of consent; or they were unaware of their diagnosis of malignancy.

### 2.2. Data Collection

A nutritional assessment was completed within seven days of surgery by site investigator dietitians using the SGA [13], and patients were classified as well-nourished (SGA A) or malnourished (SGA B or C). Further data were collected using a purpose-built data collection tool, including assessment of preoperative weight loss (per patient reported history/medical record entries) and information pertaining to receipt of preoperative dietetics (type, location, number of appointments, timing of last appointment) or nutritional intervention (type, timing, any other nutritional advice provided) (Appendix A) [12]. Clinically significant weight loss was deemed to be ≥5% in 6 months prior to surgery [14]. Percentage weight loss was also calculated at the 3-month, 1-month, and 2-week preoperative timeframes. Clinical data were obtained from participant’s medical records, including age, sex, postcode of residence to determine metropolitan or regional/rural locality, surgery type (gastric, oesophageal, or pancreatic), tumour type, pathological tumour stage from intraoperative histopathology [15], receipt of neoadjuvant chemo/radiotherapy, surgical LOS, and surgical complications (converted into a binary variable of ‘no complications’ or ‘one or more complication’) (Appendix A) [12].

### 2.3. Statistical Analysis

Statistical analyses were conducted using Stata/IC 16.0 software (StataCorp, 2020). Descriptive statistics included frequencies and percentages for categorical variables and means (standard deviation (SD)) for normally distributed continuous variables. Differences in outcomes between surgical procedures were determined using Fisher’s exact test with two-sided significance. Univariate logistic regression analysis was used to determine demographic and clinical factors associated with receiving preoperative dietetics intervention or nutrition support. Factors significant on univariate were tested with a multivariate logistic regression model to determine independent associations. For these analyses, ‘dietetics intervention’ and ‘received nutrition support’ were defined as ‘yes/no’ binary variables, with ‘unsure’ responses (*n* = 3 and *n* = 2, respectively) removed from the dataset. Previous meta-analyses have also combined preoperative modalities of nutrition support for analysis purposes [16]. Associations between preoperative dietetics intervention and weight loss 2 weeks and 1 month before surgery were investigated, as clinically meaningful changes in weight can occur within these timeframes if patients are provided with appropriate preoperative intervention, and earlier timeframes (3 months and 6 months) would likely include the pre-diagnosis period for most patients. The Kruskal–Wallis test was utilised to first determine if there were any differences in weight loss for patients receiving no dietetics intervention, or 1, 2, 3–4, or >4 appointments. As a significant difference was found for the 1-month timeframe (*p* = 0.025), and borderline significance was found for the 2-week timeframe (*p* = 0.058), pairwise analyses were undertaken for the 1-month timeframe, which determined that the differences were significant for patients receiving ≥3 appointments versus patients receiving 0–2 appointments. Therefore, the number of dietetics appointments was recoded into a binary variable (0–2 and ≥3 appointments), and an independent samples *t*-test assuming unequal variances was utilised to determine differences in preoperative weight loss. For nutrition support, weight loss 2 weeks and 1 month before surgery for patients receiving high energy high protein (HEHP) oral supplements for >2 weeks prior to surgery compared with patients who did not receive any HEHP supplements/received them for <2 weeks were investigated using independent samples t-test assuming unequal variances. Multivariate models adjusting for age, surgical procedure, tumour stage, preoperative weight loss of ≥5% in 6 months, and malnutrition were developed to determine associations between receipt of ≥3 dietetics appointments and HEHP supplements >2 weeks with LOS (continuous outcome, linear regression) and surgical complications (binary outcome, logistic regression). A sensitivity analysis using the same models and variables was also undertaken for malnourished patients only.

## 3. Results

After screening, 217 patients were eligible, and 200 consented and were included in the study. Table 1 describes the baseline clinical characteristics of participants. There were 42% pancreatic, 33% oesophageal, and 25% gastric surgeries, most of which were for adenocarcinomas (85%). Malnutrition prevalence was 42%, whilst 49% reported clinically significant weight loss.

### 3.1. Dietetics and Nutritional Intervention

Table 2 describes the dietetics and nutritional interventions patients received per surgery type. Overall, 60% of participants received preoperative dietetics intervention, whilst 50% received nutrition support. Location of dietetics care varied and was mainly performed in a chemo/radiotherapy department (41%), onsite UGI clinic (23%) or a preoperative inpatient admission (22%) (Figure 1). A higher proportion of patients undergoing oesophageal and gastric cancer surgery received dietetics intervention and nutrition support than those undergoing pancreatic surgery, and they received more appointments (Table 2, Figure 2). Furthermore, the location of dietetics intervention for pancreatic surgeries was more likely to be in hospital during a preoperative inpatient admission (Figure 2), and nutrition support was taken for a shorter timeframe prior to surgery (Table 2).

### 3.2. Factors Associated with Receipt of Dietetics Intervention and Nutrition Support

On univariate model analysis, factors associated with receipt of preoperative dietetics intervention included malnutrition and neoadjuvant therapy, whilst pancreatic surgery and associated tumour types were associated with a reduced likelihood of receiving dietetics intervention (Table 3). Factors associated with receipt of preoperative nutrition support included ≥5% weight loss prior to surgery, neoadjuvant therapy, and metropolitan residence location, whilst having pancreatic surgery was associated with a reduced likelihood of receiving preoperative nutrition support (Table 3). Factors independently associated with preoperative dietetics intervention on multivariate analysis were receipt of neoadjuvant therapy and malnutrition, whilst for preoperative nutrition support these were neoadjuvant therapy and metropolitan residence location (Table 4). Whilst malnutrition was associated with receiving preoperative dietetics intervention, 31% of malnourished participants did not receive this care, and 37% did not receive preoperative nutrition support (Table 3, Figure 3). Only 4 (5%) of the malnourished participants had a preoperative feeding tube in situ on admission for surgery.

### 3.3. Associations between Preoperative Dietetics Intervention and Nutrition Support with Preoperative Weight Loss 2 Weeks and 1 Month before Surgery

There were no statistically or clinically significant differences in percentage weight loss during the 1-month or 2-week preoperative timeframes between patients who received dietetics intervention and those who did not (*p* = 0.124 and *p* = 0.360, respectively). However, patients who received ≥3 appointments lost less weight during the 2-week (*p* = 0.022) and 1-month (*p* = 0.001) preoperative timeframes compared with patients who received 0–2 appointments (Table 5). Similarly, patients who received HEHP supplements for >2 weeks lost less weight in both timeframes compared with patients who did not (*p* = 0.007 and *p* = 0.001, respectively) (Table 5).

### 3.4. Associations between Preoperative Dietetics Intervention and Nutrition Support with Surgical Outcomes (LOS and Surgical Complications)

On multivariate analysis, there were no associations found between preoperative dietetics intervention or nutrition support with either LOS or surgical complications. When sensitivity analysis was undertaken for malnourished patients, ≥3 appointments was independently associated with reduced likelihood of complications (odds ratio 0.3, 95% confidence interval (CI) 0.1, 0.9, *p* = 0.04) (Appendix A), whilst HEHP supplements >2 weeks was associated with reduced LOS (regression coefficient −7.3, 95% CI −14.3, −0.3, *p* = 0.04) (Appendix A).

## 4. Discussion

Previous studies have called for more detailed investigation into the preoperative nutritional interventions UGI cancer patients receive in clinical practice, including frequency of dietetics contact and type of nutrition support [17,18]. Although clinicians report low access to preoperative nutrition care [19], formal data from a multi-centre study have not been previously available to identify the evidence–practice gaps. NOURISH is the first study to provide detailed, patient-level benchmarking data across oesophageal, gastric, and pancreatic surgeries, including an analysis of factors associated with receiving preoperative intervention and links with weight loss and surgical outcomes.

### 4.1. Dietetics Intervention and Nutrition Support Received

Dietitians are well placed to provide medical nutrition therapy (MNT) and individualised nutrition care to patients with UGI cancer, whilst liaising with surgeons and oncologists regarding symptom management and escalation of nutrition support. Several reviews have demonstrated that intervention by a dietitian improves nutrition-related outcomes and LOS in gastrointestinal cancer surgery [20,21,22]. Considering that guidelines recommend all cancer patients who are high risk of malnutrition receive assessment by a dietitian [23], it is concerning that 39% of the NOURISH cohort reported never having met with a dietitian before surgery. Whilst malnutrition was associated with receipt of dietetics intervention and nutrition support, over one-third of malnourished patients were not receiving any intervention. Insufficient and delayed dietetics services in gastrointestinal cancer was reported over 15 years ago [24], with the results of this study and other recent studies indicating that sufficient progress has not been made. A 2018 Victorian prevalence study of cancer patients demonstrated that of 137 patients with UGI cancer (across both palliative and curative treatment trajectories), only 40% overall and only 37% of malnourished patients were receiving dietetics intervention [25]. Recent prevalence studies of oesophageal and pancreatic cancer patients undergoing surgery in Sweden and the Netherlands report similar rates of dietetics intervention for these respective tumour groups [17,26].

The ESPEN guidelines recommend that patients undergoing UGI cancer surgery should receive preoperative oral nutrition supplements regardless of their nutritional status; however, only 30% of patients in this study reported taking HEHP supplements prior to surgery. Previous studies have reported that the presence of a dedicated dietitian is associated with increased likelihood of nutrition support prescription, and the low rates of dietetics intervention could explain the low uptake of preoperative nutrition support [27]. However interestingly, 41% of patients reported receiving preoperative nutritional advice from other health care professionals. Perhaps this could explain why metropolitan area code was associated with increased likelihood of receiving nutrition support but not with dietetics intervention. These patients may be accessing oral supplements even if they had not seen a dietitian. Prescription by other health care professionals could also explain why unintentional weight loss was associated with receiving nutrition support but not with dietetics intervention on univariate analysis. These results highlight the importance of collaboration between dietitians and the multi-disciplinary team to ensure patients receive appropriate and effective nutrition advice.

Neoadjuvant therapy remained an independent factor on multivariate analysis associated with receiving both dietetics intervention and nutrition support. This supports previous reports that there are service gaps in the outpatient setting, outside of chemotherapy or radiotherapy services [6,19]. The Swedish study also demonstrated that neoadjuvant therapy was associated with increased likelihood of receiving dietetics intervention in oesophageal cancer [17]. Pancreatic surgery was consistently associated with lower rates of dietetics service provision and nutrition support in this study, and these patients also received fewer appointments. Furthermore, patients were more likely to be seen by a dietitian during an inpatient admission than oesophageal or gastric cancers, indicating a ‘reactive’ mode of care. It is likely that the lower use of neoadjuvant chemotherapy, the shorter timeframe from diagnosis to surgery, and the clinical presentation often requiring hospitalization at diagnosis (e.g., biliary obstruction) [28] can explain some of these differences. Historically there may be inadequate funding for models of presurgical nutrition care for these patients, as only a low proportion of patients proceeded to curative surgery [26]. This group of patients is clearly highlighted as requiring service provision improvement, particularly given that weight loss has been demonstrated to impact survival post curative surgery [29]. The ESPEN guidelines highlight elderly patients as a high-risk group who should receive preoperative nutritional assessment [4]. Interestingly, age was not associated with receipt of dietetics intervention or nutrition support in this study. As age is absent from many malnutrition screening tools, it may not be taken into consideration during initial malnutrition screening processes.

The use of telehealth has significantly increased in the COVID-19 era, with significant benefits described in terms of improved access and efficiency of services [30]. Conducted prior to pandemic restrictions, this study demonstrated a very low use of telehealth, as well as community-based dietetics. These services could be utilised to assist overextended acute hospital services in reducing outpatient service gaps. Clinicians have recognised the need to consider expanding systems and workforce to improve access to nutrition care in UGI cancer, particularly in the early stages of the treatment trajectory [19]. However, barriers include appropriate expertise, timeliness of referral, and establishment of the service within the surgical oncology multi-disciplinary team. Structured nutrition care pathways can provide guidance to clinicians, reduce variations in care, and improve access to dietetics services [22], and therefore should be considered for implementation and research in UGI cancer surgery.

Enhanced Recovery After Surgery (ERAS) and ESPEN guidelines recommend carbohydrate loading prior to major UGI surgery [2,31]. Only 8% of the total cohort reported receiving carbohydrate drinks prior to surgery, indicating that translation of guidelines into practice is significantly lacking. Surprisingly, there was also a very low prevalence of enteral/parenteral feeding, as well preoperative feeding tube insertions. Given the high rates of malnutrition and recommendations for escalation of nutrition therapy [2], we would have expected a higher prevalence of preoperative enteral feeding. This could be due to the lack of evidence from high quality trials regarding enteral feeding in neoadjuvant therapy in UGI cancers. Based on several randomised controlled trials, in head and neck cancers, tumour-specific evidence-based guidelines recommend prophylactic feeding tube insertion for high-risk patients undergoing radiotherapy, and studies indicate that compliance with recommendations is as high as 80% [32]. To produce similar tumour specific evidence-based guidelines, further research regarding the use of enteral feeding during neoadjuvant therapy for UGI cancers is required.

### 4.2. Associations between Preoperative Dietetics and Nutrition Support with Clinical Outcomes

Preoperative weight loss has been associated with shorter survival in cancer patients [3]. Results of this study indicate that more intensive dietetics care and the extended use of oral nutrition supplements can reduce immediate preoperative weight loss. A previous systematic review demonstrated that intensive dietetics intervention and nutrition support can reduce weight loss and improve surgical outcomes in oesophageal cancer resection, although the evidence was considered ‘very low’ by GRADE assessment [21]. The results of this observational study provide data to support future interventional trials with regard to early and intensive preoperative dietetics intervention. Furthermore, benefits were seen in this study for malnourished patients receiving dietetics intervention and oral nutrition support with regard to LOS and complications. This is consistent with prior literature and guidelines demonstrating that malnourished patients benefit from at least 10 days of preoperative nutrition support with regard to surgical outcomes [4]. However, the optimal timing and type of preoperative nutrition intervention remains unknown. The prevalence of preoperative malnutrition and weight loss prior to surgery in this study were high, and it is unclear if the interventions provided were optimal, despite the positive results for surgical outcomes in malnourished participants. Further research is required regarding the optimal timing, type, and frequency of nutritional intervention with regard to nutritional status and treatment outcomes. Studies should also investigate impact on quality of life and physical function, which are becoming increasingly recognised in cancer survivorship. Finally, hospital costs should also be taken into consideration, as evidence-based models of nutrition care in other tumour streams have demonstrated reductions in hospital costs [33].

### 4.3. Strengths and Limitations

This was the first study to describe in detail the preoperative dietetics and nutrition intervention in a large UGI cancer surgical cohort, as well as to investigate clinical factors associated with receipt of intervention. Recruitment across 27 tertiary centres allowed for a nationwide representative sample. However, changes in nutritional status over time using the SGA were not able to be investigated due to funding limitations. There were also weight data missing for some patients. Further details of nutrition interventions (e.g., supplements prescribed, nutrition counselling provided) and compliance were also not able to be measured. Further research should prospectively investigate rates of malnutrition and nutritional interventions received from time of diagnosis to surgery.

## 5. Conclusions

Findings of this study confirm the hypothesized low rates of preoperative dietetics consultation and nutrition support in this population, which were associated with increased preoperative weight loss and risk of increased LOS and complications in malnourished patients. The results provide insights into key areas for improvement and further research to move towards optimal nutrition care prior to UGI cancer surgery. Improvements in nutrition services are required in the outpatient setting, particularly for malnourished patients and those undergoing pancreatic surgeries. Areas for further research from randomised clinical trials include the implementation of standardised preoperative nutrition care pathways and early/intensive nutrition support including enteral feeding.

## Figures and Tables

**Figure 1 nutrients-13-03205-f001:**
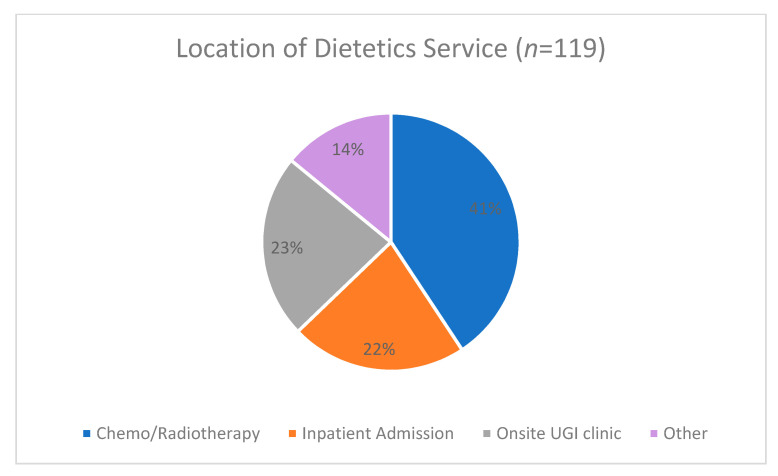
Location of dietetics service (*n* = 119). UGI, upper gastrointestinal.

**Figure 2 nutrients-13-03205-f002:**
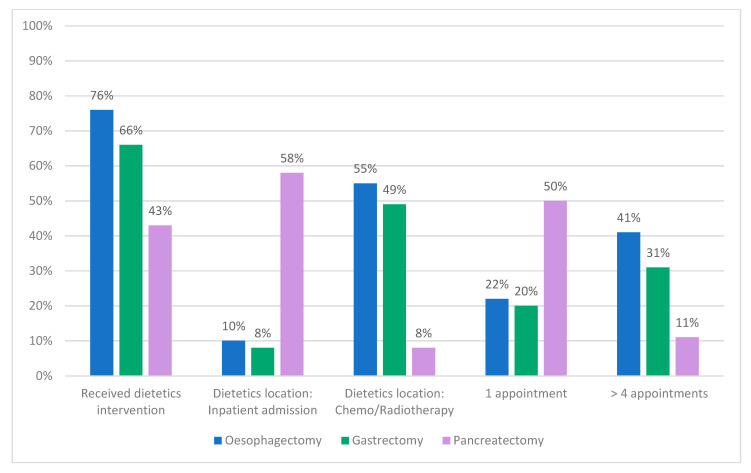
Differences between dietetics intervention received by patients undergoing oesophageal, gastric, and pancreatic surgery (*n* = 119).

**Figure 3 nutrients-13-03205-f003:**
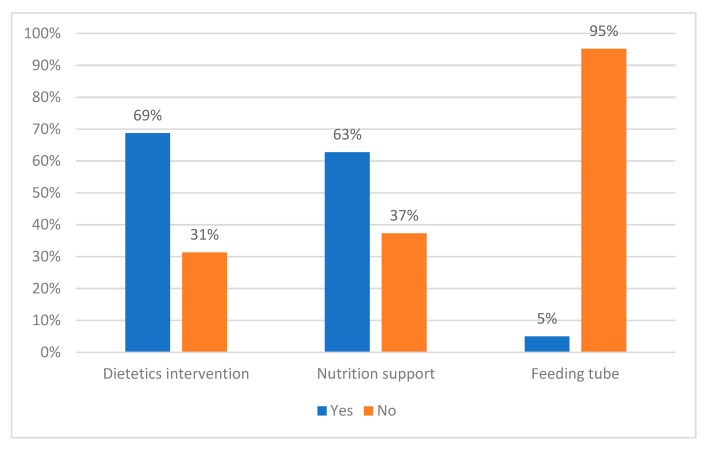
Malnourished participants (*n* = 84): receipt of preoperative dietetics intervention, nutrition support, and feeding tube present on admission to surgery.

**Table 1 nutrients-13-03205-t001:** Baseline clinical characteristics of the cohort (*n* = 200).

Variables		
Age (mean, SD ^a^)	67	10
Sex (*n*, %)		
Male	117	58.5%
Female	83	41.5%
Surgery Type (*n*, %)		
Gastric ^a^	51	25.5%
Oesophageal	66	33.0%
Pancreatic ^b^	83	41.5%
Tumour Type (*n*, %)		
Adenocarcinoma	170	85.0%
SCC	11	5.5%
GIST	2	1.0%
NET	11	5.5%
Other	6	3.0%
Intraoperative Tumour Stage (*n*, %)		
T0	15	7.5%
T1	44	22.0%
T2	49	24.5%
T3	63	31.5%
T4	14	7.0%
TX	2	1.0%
Unknown/unassessed	13	6.5%
Received Neoadjuvant Therapy (*n*, %)		
No	106	53.0%
Yes	93	47.0%
Type of Neoadjuvant therapy (*n*, %)		
Chemotherapy	52	55.0%
Chemotherapy and Radiotherapy	41	44.0%
Unknown	1	1.0%
Malnourished (*n*, %)		
No	116	58.0%
Yes	84	42.0%
≥5% LOW in 6 months ^c^ (*n*, %)		
No	95	51.0%
Yes	91	49.0%

^a^ Includes total, subtotal, partial, and distal gastrectomy. ^b^ Includes total, distal, partial, pancreatico-duodenectomy. ^c^ Expressed as a proportion of those who had 6-month weight data available (*n* = 186). SD = standard deviation, SCC = squamous cell carcinoma, GIST= gastrointestinal stromal tumour, NET = neuroendocrine tumour, LOW = loss of weight.

**Table 2 nutrients-13-03205-t002:** Dietetics and nutritional intervention by surgery type.

Variable	Overall		Oesophagectomy		Gastrectomy		Pancreatectomy		*p* Value
	*n* = 200		*n* = 66		*n* = 50		*n* = 84		
Preoperative Dietetics Intervention (*n*, %)									**<0.001**
No	78	39.0%	15	22.7%	15	30.0%	48	57.1%	
Yes	119	59.5%	50	75.8%	33	66.0%	36	42.9%	
Unsure	3	1.5%	1	1.5%	2	4.0%	0	0.0%	
Location of Preoperative Dietetics care ^a^ (*n*, %)									
Chemotherapy	44	29.7%	22	31.9%	19	48.7%	3	7.5%	**<0.001**
Radiotherapy	16	10.8%	16	23.1%	0	0.0%	0	0.0%	**<0.001**
Inpatient admission	33	22.2%	7	10.1%	3	7.7%	23	57.5%	**0.002**
Onsite UGI clinic	34	22.9%	14	20.3%	11	28.2%	9	22.5%	0.131
Onsite oncology clinic	4	2.8%	3	4.4%	1	2.6%	0	0.0%	NA
Onsite general clinic	2	1.4%	1	1.4%	1	2.6%	0	0.0%	NA
Community health	1	0.7%	0	0.0%	1	2.6%	0	0.0%	0.221
Private dietitian	0	0.0%	0	0.0%	0	0.0%	0	0.0%	NA
Preadmission clinic	8	5.4%	3	4.4%	1	2.6%	4	10.0%	0.705
Telehealth/Phonecalls	6	4.1%	3	4.4%	2	5.2%	1	2.5%	0.436
Number of appointments ^a^ (*n*, %)									**0.009**
1 appointment	36	29.5%	11	21.6%	7	20.0%	18	50.0%	
2 appointments	21	17.2%	6	11.8%	6	17.1%	9	25.0%	
3–4 appointments	20	16.4%	8	15.7%	7	20.0%	5	13.9%	
>4 appointments	36	29.5%	21	41.2%	11	31.4%	4	11.1%	
Unsure	9	7.4%	5	9.8%	4	11.5%	0	0.0%	
Timing of last appointment ^a^ (*n*, %)									0.252
1–2 weeks before surgery	63	51.6%	25	49.0%	17	48.6%	21	58.3%	
2–4 weeks before surgery	28	23.0%	11	21.6%	7	20.0%	10	27.8%	
>1 month before surgery	23	18.9%	13	25.5%	7	20.0%	3	8.3%	
>3 months before surgery	3	2.5%	0	0.0%	1	2.9%	2	5.6%	
Unsure	5	4.1%	2	3.9%	3	8.6%	0	0.0%	
Prior nutritional advice from other HCP (*n*, %)									**0.006**
No	94	47.0%	21	31.8%	24	48.0%	49	58.3%	
Yes	81	40.5%	32	48.5%	18	36.0%	31	36.9%	
Unsure	25	12.5%	13	19.7%	8	16.0%	4	4.8%	
Type of advice from other HCP ^b^ (*n*, %)									0.561
Advice so they can gain weight	15	14.2%	7	15.6%	3	11.5%	5	14.3%	
Advice so they can lose weight	11	10.4%	3	6.7%	3	11.5%	5	14.3%	
High protein	21	19.8%	11	24.4%	5	19.2%	5	14.3%	
Nutritional supplement drinks	22	20.8%	8	17.8%	4	15.4%	10	28.6%	
Other	10	9.4%	2	4.4%	3	11.5%	5	14.3%	
Unsure	27	25.5%	14	31.1%	8	30.8%	5	14.3%	
Preoperative Nutrition Support (*n*, %)									**0.033**
No	98	49.0%	27	40.9%	20	40.0%	51	60.7%	
Yes	99	49.5%	38	57.6%	28	56.0%	33	39.3%	
Unsure	3	1.5%	1	1.5%	2	4.0%	0	0.0%	
Type of Preoperative Nutrition Support ^c^ (*n*, %)									
HEHP drinks	72	63.2%	31	67.4%	20	64.5%	21	56.8%	**0.017**
Immunonutrition	21	18.4%	5	10.9%	6	19.4%	10	27.0%	0.639
Carbohydrate loading	9	7.9%	5	10.8%	0	0.0%	4	10.8%	0.148
Enteral Nutrition	7	6.1%	4	8.7%	2	6.5%	1	2.7%	0.266
Parenteral Nutrition	2	1.8%	0	0.0%	1	3.2%	1	2.7%	0.548
Unsure	3	2.6%	1	2.2%	2	6.5%	0	0.0%	0.183
If on HEHP drinks, length of prescription (*n*, %) ^d^									**0.019**
5 days before surgery	8	10.3%	1	3.1%	3	12.5%	4	18.2%	
1 week before surgery	9	11.5%	1	3.1%	4	16.7%	4	18.2%	
>2 weeks before surgery	13	16.7%	5	15.6%	2	8.3%	6	27.3%	
>1 month before surgery	23	29.5%	11	34.4%	5	20.8%	7	31.8%	
>3 months before surgery	23	29.5%	14	43.8%	8	33.3%	1	4.5%	
Unsure	2	2.6%	0	0.0%	2	8.3%	0	0.0%	
If on Immunonutrition drinks, length of prescription ^e^ (*n*, %)									NA
5 days before surgery	21	100%	5	100%	6	100%	10	100%	
1 week before surgery	0	0.0%	0	0.0%	0	0.0%	0	0.0%	
>2 weeks before surgery	0	0.0%	0	0.0%	0	0.0%	0	0.0%	
>1 month before surgery	0	0.0%	0	0.0%	0	0.0%	0	0.0%	
>3 months before surgery	0	0.0%	0	0.0%	0	0.0%	0	0.0%	
Feeding tube present on admission for surgery (*n*, %)									**<0.001**
No	195	97.5%	63	95.5%	49	98.0%	83	99.0%	
Yes	5	2.5%	3	4.5%	1	2.0%	1	1.0%	

^a^ Expressed as a proportion of those who responded ‘yes/unsure’ to preoperative dietetics intervention (*n* = 122, patients could select multiple options). ^b^ Expressed as a proportion of those who responded ‘yes/unsure’ to receiving advice from other HCPs (*n* = 106, patients could select multiple options). ^c^ Expressed as a proportion of those who responded ‘yes’ to preoperative nutrition support (*n* = 99, patients could select multiple options). ^d^ Expressed as a proportion of those who responded ‘yes’ to HEHP drinks (*n* = 62). ^e^ Expressed as a proportion of those who responded ‘yes’ to preoperative immunonutrition (*n* = 21). HCP = health care professional, HEHP = high energy high protein, NA= not applicable. Bolded *p* values indicate statistical significance.

**Table 3 nutrients-13-03205-t003:** Univariate analysis of factors associated with receiving dietetics intervention and preoperative nutrition support.

	Dietetics Intervention					Preoperative Nutrition Support				
	Bivariate Analysis			Univariate Logistic Model		Bivariate Analysis			Univariate Logistic Model Analysis	
Variable	Did Not Receive Dietetics Intervention *n* (%)	Received Dietetics Intervention *n* (%)	*p* Value	Odds Ratio (95% CI)	*p* Value	Did Not Receive Nutrition Support *n* (%)	Received Nutrition Support *n* (%)	*p* Value	Odds Ratio (95% CI)	*p* Value
Age			0.883					0.562		
<65	31 (40.3)	46 (59.7)		1.0 (Ref)		41 (52.6)	37 (47.4)		1.0 (Ref)	
≥65	47 (39.2)	73 (60.8)		1.1 (0.6, 1.9)	0.878	57 (47.9)	62 (52.1)		1.3 (0.7, 2.2)	0.420
Sex (*n*, %)			0.557					0.885		
Male	43 (37.7)	71 (62.3)		1.0 (Ref)		58 (50.4)	57 (49.6)		1.0 (Ref)	
Female	35 (42.2)	48 (57.8)		0.8 (0.5, 1.5)	0.529	40 (48.8)	42 (51.2)		1.1 (0.6, 1.9)	0.847
Surgery Type			**<0.001**					0.029		
Oesophagectomy	15 (23.1)	50 (76.9)		1.0 (Ref)		27 (41.5)	38 (58.5)		1.0 (Ref)	
Gastrectomy	15 (31.3)	33 (68.8)		0.7 (0.3, 1.5)	0.332	20 (41.7)	28 (58.3)		1.0 (0.5, 2.2)	0.921
Pancreatectomy	48 (57.1)	36 (42.9)		0.2 (0.1, 0.5)	**<0.001**	51 (60.7)	33 (39.3)		0.5 (0.2, 0.9)	0.017
Tumour Stage			**0.011**					0.375		
T0	2 (14.3)	12 (85.7)		1.0 (Ref)		5 (35.7)	9 (64.3)		1.0 (Ref)	
T1	19 (44.2)	24 (55.8)		0.2 (0.1, 1.1)	**0.058**	24 (57.1)	18 (42.9)		0.4 (0.1, 1.4)	0.162
T2	26 (53.1)	23 (46.9)		0.2 (0.1, 0.7)	**0.019**	19 (38.8)	30 (61.2)		0.8 (0.2, 2.7)	0.704
T3	16 (25.4)	47 (74.6)		0.5 (0.1, 2.5)	0.382	32 (50.8)	31 (49.2)		0.5 (0.2, 1.6)	0.229
T4	5 (35.7)	9 (64.3)		0.3 (0.1, 1.9)	0.203	6 (42.9)	8 (57.1)		0.7 (0.2, 3.0)	0.598
Tumour Location								0.160		
Bile Duct	3 (50%)	3 (50%)	**0.001**	1.0 (Ref)		3 (50.0)	3 (50.0)		1.0 (Ref)	
Gastric	15 (31.3)	33 (68.8)		2.2 (0.4, 12.2)	0.367	18 (37.5)	30 (62.5)		1.8 (0.3, 9.7)	0.507
Oesophageal	15 (25.4)	44 (74.6)		2.9 (0.5, 16.1)	0.216	25 (42.4)	34 (57.6)		1.4 (0.3, 7.5)	0.695
Pancreatic	34 (61.8)	21 (38.2)		0.6 (0.1, 3.4)	0.576	34 (61.8)	21 (38.2)		0.6 (0.1, 3.4)	0.576
Ampullary	9 (52.9)	8 (47.1)		0.9 (0.1, 5.7)	0.901	11 (64.7)	6 (35.3)		0.6 (0.1, 3.6)	0.528
Duodenal	2 (40.0)	3 (60.0)		1.5 (0.1, 16.5)	0.741	2 (60.0)	2 (40.0)		0.7 (0.1, 7.4)	0.741
GOJ	0 (0)	7 (100)		1.0 (empty)		4 (57.1)	3 (42.9)		0.8 (0.1, 6.7)	0.797
Tumour Type			0.429					0.292		
Adenocarcinoma	64 (38.3)	103 (61.7)		1.6 (0.3, 8.2)	0.567	83 (49.4)	85 (50.6)		0.5 (0.1, 2.9)	0.463
SCC	3 (27.3)	8 (72.7)		2.7 (0.3, 21.3)	0.355	5 (50.0)	5 (50.0)		0.6 (0.1, 4.8)	0.629
GIST	1 (50.0)	1 (50.0)		1.0 (0.1, 24.5)	1.00	0	2 (100)		1.0 (empty)	
NET	7 (63.6)	4 (36.4)		0.6 (0.1, 4.3)	0.857	8 (72.7)	3 (27.3)		0.2 (0.1, 1.6)	0.128
Other	3 (50.0)	3 (50.0)		1.0 (Ref)		2 (33.3)	4 (66.7)		1.0 (Ref)	
Received Neoadjuvant Therapy			**<0.001**					**<0.001**		
No	59 (55.7)	47 (55.3)		1.0 (Ref)		66 (62.9)	39 (37.1)		1.0 (Ref)	
Yes	19 (20.9)	71 (79.1)		4.8 (2.5, 9.0)	**<0.001**	32 (34.8)	60 (65.2)		3.2 (1.8, 5.6)	**<0.001**
Location of Residence			0.748					**0.008**		
Rural/Regional	24 (42.1)	33 (57.9)		1.0 (Ref)		37 (64.9)	20 (35.1)		1.0 (Ref)	
Metropolitan	54 (38.6)	86 (61.4)		1.2 (0.6, 2.2)	0.646	61 (43.6)	79 (56.4)		2.5 (1.3, 4.7)	**0.005**
Nutritional Status			**0.055**					**0.004**		
Well Nourished	52 (45.6)	62 (54.4)		1.0 (Ref)		67 (58.8)	47 (41.2)		1.0 (Ref)	
Malnourished	26 (31.3)	57 (68.7)		1.8 (1.1, 3.3)	**0.044**	31(37.3)	52 (62.7)		2.3 (1.3, 4.2)	**0.004**
≥5% LOW in 6 months			0.444					**0.011**		
No	44 (41.1)	63 (58.9)		1.0 (Ref)		61 (57.0)	46 (43.0)		1.0 (Ref)	
Yes	27 (34.6)	51 (65.4)		1.3 (0.7, 2.4)	0.369	29 (37.2)	49 (62.8)		2.2 (1.2, 4.0)	**0.010**

OR = odds ratio, CI = confidence interval, LOW = loss of weight, GOJ = gastro-oesophageal, SCC = squamous cell carcinoma, GIST = gastrointestinal stromal tumour, NET = neuroendocrine tumour. Bolded *p* values indicate statistical significance.

**Table 4 nutrients-13-03205-t004:** Factors independently associated with receiving preoperative dietetics intervention or nutrition support by multivariate analysis.

Variable	Dietetics Intervention OR (95% CI)	*p* Value	Nutrition Support OR (95% CI)	*p* Value
Surgical Procedure				
Gastrectomy	0.6 (0.2, 1.6)	0.299	1.2 (0.5, 2.7)	0.731
Pancreatectomy	0.4 (0.2, 1.3)	0.144	0.9 (0.4, 2.2)	0.788
Tumour Stage				
T1	0.5 (0.1, 2.9)	0.463		
T2	0.4 (0.1, 2.2)	0.289		
T3	1.0 (0.2, 5.8)	0.975		
T4	0.6(0.1, 4,9)	0.651		
Received Neoadjuvant Therapy	3.6 (1.6, 8.1)	**0.003**	3.3 (1.5, 7.4)	**0.004**
Malnourished	2.1 (1.1, 4.3)	**0.046**	2.0 (0.8, 4.9)	0.120
Metropolitan Residence			2.6 (1.3, 5.4)	**0.008**
≥5% LOW in 6 months			1.4 (0.6, 3.3)	0.460

OR = odds ratio, CI = confidence interval, LOW = loss of weight. Bolded *p* values indicate statistical significance.

**Table 5 nutrients-13-03205-t005:** Associations between preoperative dietetics intervention and HEHP supplements with preoperative weight loss 2 weeks and 1 month before surgery.

Timeframe before Surgery	0–2 Dietetics Appts	≥3 Dietetics Appts	Mean %^b^ Difference (95% CI)	*p* Value	No HEHP HEHP <2 Weeks	HEHP > 2 Weeks	Mean % Difference (95% CI)	*p* Value
*N*	65	42			68	44		
2 weeks % LOW ^a^	1.6 (2.4)	0.7 (1.7)	0.90 (0.1, 1.7)	**0.022**	1.6 (2.3)	0.6 (1.5)	1.00 (0.2–1.8)	**0.007**
*N*	63	39			63	45		
1 month % LOW ^a^	3.1 (3.3)	1.2 (2.0)	1.8 (0.8, 2.9)	**0.001**	2.9 (3.4)	1.2 (1.8)	1.7 (0.8, 2.7)	**0.001**

^a^ Results presented as mean (standard deviation). ^b^ Percent. Appts = appointments, HEHP = high energy high protein, LOW = loss of weight. CI = confidence interval. Bolded *p* values indicate statistical significance.

## Data Availability

Not applicable.

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
