# Peer review of "Preoperative Nutrition Intervention in Patients Undergoing Resection for Upper Gastrointestinal Cancer: Results from the Multi-Centre NOURISH Point Prevalence Study"

_nutrients, 2021, doi:10.3390/nu13093205_

Round 1

Reviewer 1 Report

A very well designed and carried out study, provides evidence-based  guidelines for clinical practice.

Author Response

Many thanks for your review. 

Reviewer 2 Report

This top trial shows the importance of the intensity of nutritional intervention in the preoperative setting (rather than examining only one single nutrient as a “golden bullet”), resulting in the improvement of practical clinical endpoints. Moreover, it demonstrates that many malnourished patients are still being insufficiently treated. I have the following comments:

  • The prevalence and incidence of malnutrition appears to be quite low for GI cancer, as numbers as high as 85% have been reported earlier. Is this a special subgroup overall (education, income,…)?
  • The number of patients receiving enteral and/or parenteral nutrition, appears to be nearly negligible, along with the very low number of patients having a preoperative feeding tube. Although indeed not being the first-line therapy of choice, this is somewhat lower than reported, and the authors should comment on this.
  • Was the dietetic intervention continued postoperatively, or did the patients switch to a regular programme?

Author Response

Reviewer 2:

This top trial shows the importance of the intensity of nutritional intervention in the preoperative setting (rather than examining only one single nutrient as a “golden bullet”), resulting in the improvement of practical clinical endpoints. Moreover, it demonstrates that many malnourished patients are still being insufficiently treated. I have the following comments:

Many thanks for your review, we have addressed the comments below.

The prevalence and incidence of malnutrition appears to be quite low for GI cancer, as numbers as high as 85% have been reported earlier. Is this a special subgroup overall (education, income,…)?

This is a worthwhile detail to note. Although we agree that previous early research has demonstrated malnutrition prevalence as high as 85% or higher in some early studies. However for this study the rates of malnutrition in this study were also consistent with several studies in UGI cancer including:

[14] Marshall KM, Loeliger J, Nolte L, Kelaart A, Kiss NK. Prevalence of malnutrition and impact on clinical outcomes in cancer services: A comparison of two time points. Clin Nutr. 2019;38:644-51.

[24] Garth AK, Newsome CM, Simmance N, Crowe TC. Nutritional status, nutrition practices and post-operative complications in patients with gastrointestinal cancer. J Hum Nutr Diet. 2010;23:393-401.

[25] Hill A, Kiss N, Hodgson B, Crowe TC, Walsh AD. Associations between nutritional status, weight loss, radiotherapy treatment toxicity and treatment outcomes in gastrointestinal cancer patients. Clin Nutr. 2011;30:92-8.

It is also noted that his is a curative surgical cohort which may have less weight loss than patients with more advanced cancers undergoing palliative treatment.

We have not included this information in the manuscript as we have not focused on malnutrition prevalence in the paper, but rather as a baseline characteristic- however if you would like us to include this information please let us know.

The number of patients receiving enteral and/or parenteral nutrition, appears to be nearly negligible, along with the very low number of patients having a preoperative feeding tube. Although indeed not being the first-line therapy of choice, this is somewhat lower than reported, and the authors should comment on this.

Yes, we agree this was surprising. This has been commented on in the discussion but we have added further discussion as to the surprising nature of these findings especially given the guidelines for escalation of nutrition support, lines 319-323:

“Surprisingly, there was also a very low prevalence of enteral/parenteral feeding, as well preoperative feeding tube insertions. Given the high rates of malnutrition and recommendations for escalation of nutrition therapy [2], we would have expected higher prevalence of preoperative enteral feeding. This could be due to the lack of evidence from high quality trials regarding enteral feeding in neoadjuvant therapy in UGI cancers. In head and neck cancers, tumour specific evidence-based guidelines recommend prophylactic feeding tube insertion for high risk patients undergoing radiotherapy based on several randomised controlled trials, and studies indicate compliance with recommendations is as high as 80% [32]. Further research regarding the use of enteral feeding during neoadjuvant therapy for UGI cancers is required, to produce similar tumour specific evidence-based guidelines.”

Was the dietetic intervention continued postoperatively, or did the patients switch to a regular programme?

The postoperative dietetic management of patients included in this study will be reported in an additional manuscript. As this was a prevalence study, the management of patients was not set but initiated as per the treating dietitian/surgical team at each facility.

Reviewer 3 Report

well written and I did not find points for revision or correction

Author Response

Many thanks for your review.